# Diversity and dynamics of bacteria from iron-rich microbial mats and colonizers in the Mediterranean Sea (EMSO-Western Ligurian Sea Observatory): Focus on Zetaproteobacteria

Aina Astorch-Cardona[1], Lionel Bertaux[2], Yann Denis[3], Alain Dolla[1], Céline Rommevaux[1]*

1 Aix Marseille Univ., Université de Toulon, CNRS, IRD, MIO, Marseille, France, 2 Aix Marseille Univ., CNRS, LCB, Marseille, France, 3 Institut de Microbiologie de la Méditerranée, CNRS - Aix Marseille Université, Marseille, France

☯ These authors contributed equally to this work.

* celine.rommevaux@mio.osupytheas.fr

**Data Availability Statement:** The data sets generated and analyzed during the current study are publicly available. The data can be found here:

## Abstract

Autotrophic microaerophilic iron-oxidizing Zetaproteobacteria seem to play an important role in mineral weathering and metal corrosion in different environments. Here, we compare the bacterial and zetaproteobacterial communities of a mature iron-rich mat together with *in situ* incubations of different Fe-bearing materials at the EMSO-Ligure West seafloor observatory, which is located on the abyssal plain in the NW Mediterranean Sea. Our results on bacterial communities enable us to make a clear distinction between those growing on mild steel anthropic substrata and those developing on basaltic substrata. Moreover, on anthropic substrata we highlight an influence of mat age on the bacterial communities. Regarding zetaproteobacterial communities, our results point to an increase in ZetaOTUs abundance and diversification with the age of the mat. We corroborate the key role of the ZetaOTU 2 in mat construction, whatever the environment, the substrata on which they develop or the age of the mat. We also show that ZetaOTU 28 is specific to anthropogenic substrata. Finally, we demonstrate the advantage of using dPCR to precisely quantify very low abundant targets, as Zetaproteobacteria on our colonizers. Our study, also, allows to enrich our knowledge on the biogeography of Zetaproteobacteria, by adding new information on this class and their role in the Mediterranean Sea.

## Introduction

Iron-oxidizing Zetaproteobacteria are essential in the development of iron-rich microbial mats in hydrothermal systems [1, 2]. Nonetheless, hydrothermal environments are not the only habitats where members of this class can thrive. Indeed, they have been detected from

NCBI, PRJNA1063670, the accession numbers for the BioSamples are SRR27489633 - SRR27489647. During the reviewing process they are available for reviewers by following the link: https://dataview.ncbi.nlm.nih.gov/object/ PRJNA1063670?reviewer= hn6vh1nq0nueuonqdki4ufq56f.

**Funding:** Flotte Océanographique Française (FOF) Grant EMSO-LO cruises 2018, 2022 Agence Nationale de la Recherche (ANR) Grant ANR-21-CE02-0012 Ministère de l'Education Nationale, de l'Enseignement Superieur et de la Recherche (MESR) Grant Astorch-Cardona PhD scholarship EC European Regional Development Fund(ERDF) grant 1166-39417.

**Competing interests:** The authors have declared that no competing interests exist.

deep-sea sediments to coastal habitats and even in terrestrial hot springs, across different salinities and dissolved Fe(II) (dFe) concentrations [3–5]. Besides their unanticipated broad distribution in aquatic environments, Zetaproteobacteria seem to play an important role in mineral weathering and metal corrosion, using the mineralogical structure and composition of the substratum in which they develop for their metabolism. Both *in situ* and *in vitro* incubation experiments have enabled the study of these processes in marine ecosystems, providing evidence that iron-oxidizing bacteria can potentially colonize Fe-bearing and other materials in iron-rich marine environments [6–9].

In 1993, the concept that neutrophilic iron-oxidizing bacteria could be involved in mineral weathering processes in marine environments was proposed through an analysis of microbial carbon fixation in seafloor massive sulphide deposits in hydrothermal vents at the Mid-Atlantic_Ridge (MAR) [10]. Afterwards, Zetaproteobacteria were detected in *in situ* incubations of a sulphide chimney sample at the Juan de Fuca Ridge, which hosted iron-oxidizer biofilms in its pyrrhotite-rich regions [11, 12]. Based on this study, *in situ* incubations of pyrrhotite were performed in shallow waters at Santa Catalina Island, USA, which revealed the presence of Zetaproteobacteria and other novel iron-oxidizers (*i.e.* Thiomicrospira) within them [13]. The hypothesis that basaltic glass could support the growth of iron-oxidizing bacteria was first tested in *in vitro* experiments, when these bacteria were isolated from deep-sea, low-temperature weathering deposits from the vicinities of the Juan de Fuca hydrothermal area [14]. Finally, the first direct evidence that Zetaproteobacteria could use the structural Fe(II) from basaltic glass as their sole energy source in a non-hydrothermal environment was given by both *in situ* and *in vitro* experiments using reduced basaltic glass in an abyssal plain off the MAR [15].

On the other hand, mild steel is the most typical kind of steel used for the development of large-scale infrastructures. As it is mostly composed of iron and is easily corroded, it was hypothesized that this type of steel (or other Fe-bearing metals) could support the growth of neutrophilic iron-oxidizing bacteria [16, 17], as had already been proven for Fe-bearing minerals. The first experiment to study Zetaproteobacteria in the context of microbially influenced corrosion was performed in coastal waters in Maine, USA [16]. Performing *in situ* incubations with mild steel coupons, McBeth et al. [16] showed that mild steel can be colonized by Zetaproteobacteria. Since then, several experiments focusing in mild steel colonization by members of this class have been performed, many of which centered in investigating colonization patterns over relatively short time-periods [13, 18–20]. Such studies revealed that Zetaproteobacteria might be early colonizers, as their presence decreases throughout weeks, and that anaerobic microorganisms predominate when the environment becomes more reducing. On the other hand, analyses of rusticles and corrosion tubercles (*i.e.* the products of long-term corrosion) formed on ship wrecks [21, 22] or on mild steel present for eight years in coastal waters [23], showed that Zetaproteobacteria are also present within well-established microbially induced corrosion communities. Even though there does not exist direct evidence that iron-oxidizing bacteria pit steel surfaces or accelerate microbially influenced corrosion [24, 25], they seem to be able to use Fe(II) released from steel surfaces as an energy source [16]. Emerson [17] proposed a model of how iron-oxidizing bacteria may colonize steel surfaces based on data from Mumford et al. [25]. According to this model, these bacteria colonize steel surfaces early and act as ecological engineers in them, providing an optimal environment for the further development of biofilms.

Even though several *in vitro* and *in situ* incubation studies have been performed with different Fe-bearing substrata both in the short- and long-term, less is known about the communities present in iron-rich microbial mats that develop on anthropic steel structures deployed in the environment. At approximately 2400 m depth, the steel structure of the

European Multidisciplinary Seafloor and water column Observatory- Western Ligurian Sea (EMSO-LO) has been lying in the deep coastal plain of the Mediterranean Sea for more than 10 years. EMSO-LO is a cabled seafloor observatory deployed in the NW Mediterranean Sea. Contrary to the high dissolved iron (dFe) concentration found in hydrothermal systems, *i.e.* between 185 and 2800 µM at the Lucky Strike Hydrothermal Field (LSHF) [26], the dFe concentration on the abyssal plain at the NW Mediterranean Sea is of around 0.3–0.5 nM [27]. Some years after the deployment of this observatory and within the dives performed to maintain its structure, it was observed that some of its parts were starting to rust, forming what looked like an iron-rich microbial mat. Despite the low dFe concentration detected on the abyssal plain, Zetaproteobacteria seem to be able to thrive in this environment, using the metallic structure of this observatory as a substratum for their growth and for the development of iron-rich microbial mats. These mats, together with *in situ* incubations of Fe-materials by using microbial colonizers of various compositions and during different time periods, are the focus of this study. Here, we investigate through 16S rRNA metabarcoding and digital PCR how substratum type and origin, but also incubation time, affect both the bacterial and zetaproteobacterial communities in a non-hydrothermal, biocorrosion- and mineral weathering-related context.

## Materials and methods

### Site and sample description

The study was carried out in the EMSO-LO area. EMSO-LO is one of the few deep-sea cabled observatories in the world, located at approximately 2400 m depth in the North-Western Mediterranean Sea, 42 km off the coast of Toulon, France. It is composed of two instrumented hubs dedicated to environmental studies: (i) the Module Interface Instrumented (MII) linked to a standalone deep-sea mooring dedicated to the long-term monitoring of hydrological and biogeochemical properties (Autonomous Line with a Broad Acoustic Transmission for Research in Oceanography and Sea Sciences—ALBATROSS) and (ii) the Secondary Junction Box (SJB), linked with different environmental instruments. In 2008, the SJB was initially connected to the Astronomy with a Neutrino Telescope and Abyss Environmental Research (ANTARES, 42˚48'00.0"N 6˚10'01.2"E) site [28]. Nonetheless, in 2018, the system was recovered and refurbished, and in 2019–2020 it was re-deployed at the Mediterranean Eurocentre for Underwater Sciences and Technology (MEUST, 42˚48'05.4"N 5˚58'57.0"E) site [29]. At 2400 m depth in the coastal abyssal plain of the Mediterranean Sea, the EMSO-LO area is impacted by the Northern Current (Liguro-Provençal-Catalan Current) [30, 31], affected by the recurrent formation of deep water in the Gulf of Lion [32, 33]. Nevertheless, some environmental parameters remain broadly stable over time, such as a high seawater temperature (around 13ºC) and a low dFe concentration (around 0.3–0.5 nM) [27]. However, $dO_2$ concentrations seem to be slowly diminishing over time, from ~192 µmol/ml in 2014 to ~187 µmol/ml in 2016 [34].

### Sample collection and *in situ* incubations

Iron-rich microbial mats had developed on the mild steel part of the SJB structure, while it was still connected to ANTARES, corroding it and creating rust (Fig 1). The iron-rich microbial mat sample (hereafter called FeOx EMLIG 18) was collected during the EMSO Ligure Ouest 2018 (EMSO-LO 18) cruise [35]. Sample collection was performed on board of the R.V. *Pourquoi Pas?* with the Human Operated Vehicle (HOV) *Nautile*. On the SJB base structure, the iron-rich mat covered a surface of about 500 cm². Between two and four grabs of the mat were collected using the grabber of the submersible's hydraulic arm (18 x 16 x 16 cm) and placed in

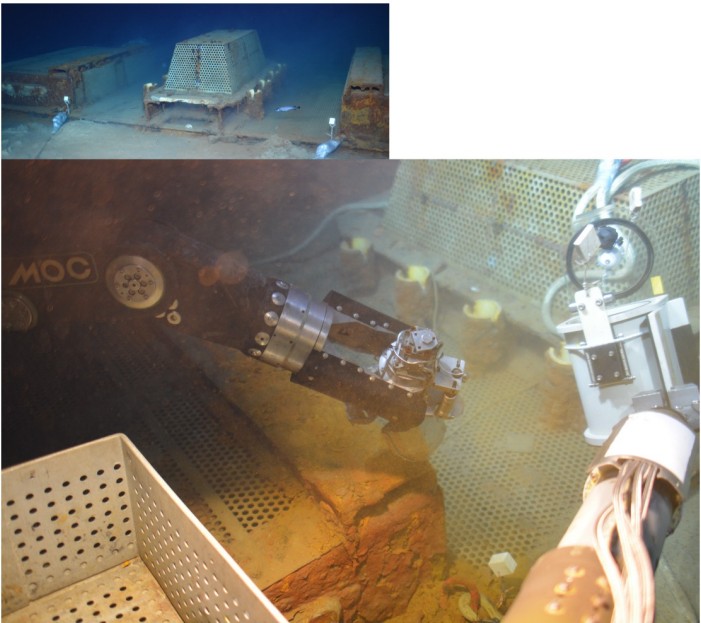

**Fig 1. Iron-rich microbial mat image.** General view of the SJB structure at the ANTARES site (top left) and image of the iron-rich microbial mat sampled with the *Nautile* hydraulic arm from the SJB structure (bottom). Images taken by the HD camera of *Nautile* during the EMSO-LO 18 cruise.

a previously sterilized bio-box to prevent sample contamination and leaching during ascent to the surface. Once onboard, it was transferred under a laminar flow hood, sterilely aliquoted in 5 ml tubes and preserved at -80˚C for various on-shore laboratory analyses.

Geomicrobiology colonizers were used to perform *in situ* incubations using different substrata and incubation times in the vicinities of EMSO-LO, both in ANTARES and MEUST sites (Fig 2A). Geomicrobiological colonization modules and microbial colonizers were made up as described by Henri et al. [15] (Fig 2B). Four different microbial colonizers were used for this study, containing as substratum, either synthetic reduced basaltic glass enriched in Fe(II) (BH2), natural basaltic glass (Bnat) recovered from the LSHF, or mild steel grit (Gr) used as

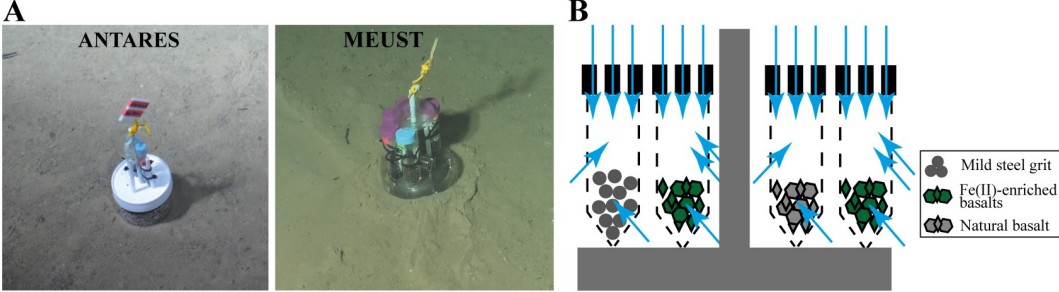

**Fig 2. Image and schema of colonizers.** A) Images of the colonizers deployed on the interphase between sediments and seawater at ANTARES site (left) containing Bnat EMLIG 20 and BH2 EMLIG 20 samples, and the one deployed on the sediments at MEUST site (right) containing BH2 EMLIG 22 and Gr EMLIG 22 samples. Images taken by the HD camera of *Nautile*, during the EMSO-LO 18 and ESSNaut 22 cruises, respectively. B) Schematic representation of microbial colonizers, showing the different substrata and the circulation of environmental seawater inside colonizers. Dark grey bars correspond to the colonizer support, blue arrows indicate the flow circulation, tubes containing substrata are in dashed lines.

HOV's *Nautile* ballast. The synthesis of the BH2 was performed from natural basaltic glass recovered from the LSHF at the Laboratoire de Géomatériaux et Environnement de l'Université Paris Est (Marne-la-Vallée, France) [15]. The synthesis consisted on a first step of re-melting the natural basalt in a non-vertical oven at an ambient atmosphere, and on a second re-melting step in a vertical oven under a reducing atmosphere ($H_2$) in order to enrich it in Fe (II), following the protocols described in Henri et al. [15].

A geomicrobiological colonization module with a microbial colonizer containing BH2 and another containing Bnat was deployed near a ballast chain at ANTARES site in January 2018, during the EMSO-LO 18 cruise [35], on board the R.V. *Pourquoi Pas*? and with the HOV *Nautile*. This geomicrobiological colonization module was recovered in November 2020, during the NIOZ-ANTARES cruise [36], on board of the R.V. *Pourquoi Pas*? with the ROV *Victor 6000*. These samples are named BH2 EMLIG 20 and Bnat EMLIG 20 throughout the text and their *in situ* incubation time was of approximately three years. Another geomicrobiological colonization module with a microbial colonizer containing BH2 and a microbial colonizer containing Gr was deployed in February 2022 at MEUST site during the EMSO Ligure Ouest 2022 (EMSO-LO 22) cruise [37]. This colonization module was recovered in December 2022, during the *Nautile* testing (ESSNaut 22) cruise [38]. Both deployment and recovery of this geomicrobiological colonization module were performed on board of the R.V. *Pourquoi Pas*? with the HOV *Nautile*. These samples are named BH2 EMLIG 22 and Gr EMLIG 22 throughout the text and their *in situ* incubation time was of approximately one year. Preparation, sterilization, deployment and recovery of the geomicrobiological colonization modules were performed as described in Henri et al. [15]. They were incubated *in situ* at the interphase between seawater and sediments at both sites of the EMSO-LO observatory (Fig 2). Once recovered on board, microbial colonizers were transferred under a laminar flow hood, and the substratum was sterilely aliquoted in 2 ml tubes and preserved at -80˚C for various on-shore laboratory analyses.

## DNA extraction

Total genomic DNA was extracted using the DNeasy® PowerSoil® Pro Kit (QIAGEN, Hilden, Germany), following the manufacturer's instructions. Cell lysis was performed using the FastPrep® Instrument (MP Biomedicals, Santa Ana, CA, USA). DNA was extracted in triplicate from the FeOx EMLIG 18 iron-rich microbial mat sample and from the BH2 EMLIG 20 microbial colonizer. DNA from the other microbial colonizers (Bnat EMLIG 20, BH2 EMLIG 22 and Gr EMLIG 22) was extracted nine times, in order to have enough material for 16S rRNA gene sequencing and digital PCR (dPCR). The extraction products of these last samples were pooled by threes before proceeding with further analyses, allowing to perform 16S rRNA gene sequencing and dPCR in the same triplicates for each sample. DNA quantification was performed using the Invitrogen™ Qubit™ 1X dsDNA High Sensitivity (HS) Assay Kit (Invitrogen Thermo Fisher Scientific) and the Qubit fluorometer (Invitrogen Thermo Fisher Scientific).

## Digital PCR

dPCR was used to perform an absolute quantification of the 16S rRNA gene for Bacteria and Zetaproteobacteria on each triplicate of all samples. dPCR was performed at the Plateforme Transcriptomique de l'Institut de Microbiologie de la Méditerranée (IMM, Marseille, France) on the Naica® System for Crystal Digital PCR™ using Sapphire Chips (Stilla, France). Each dPCR reaction mixture contained 9.375 μl of PerfeCTa® qPCR ToughMix®, UNG 2X (QuantaBio, Massachusets, USA), 1 μl of Alexa Fluor® 647 (Invitrogen Thermo Fisher Scientific) at

0.02 mg/ml, 1.9 µl of EvaGreen® Dye 20X in water (Biotium, California, USA), 0.25 µl of primers at a concentration of 20 µM, DNase-free water and 10 µl of template DNA in a final volume of 25 µl. For each dPCR reaction mixture, a DNase-free water negative control was prepared. The primer couples used for Bacteria and Zetaproteobacteria were recovered from previous literature (S1 Table).

dPCR amplification was performed on the Naica® System's thermocycler (Geode). The dPCR programs consisted of an initial 3-minute denaturation step at 95˚C, followed by 45 or 75 cycles of a 10-second denaturation step at 95˚C and a 15-second annealing step at different hybridization temperatures for each primer set (S1 Table). After amplification, an unpacking program consisting of a pressure rise up to 50 mbar (at a 20 mbar/second rate) was run five times to ensure that different individual drops were not in contact with one another, which would bias the quantification.

Fluorescence measurement was done on the Naica® System's 3-color fluorescence reader (Prism). The Crystal Reader and Crystal Miner softwares (Stilla, France) were used for experimental parameter set-up and data analysis, respectively.

All samples were diluted 1/10 for Bacteria, while for Zetaproteobacteria, 10 µl of non-diluted sample were used for each triplicate of the FeOx EMLIG 18 sample and 30 µl of non-diluted sample were used for each triplicate of each microbial colonizer. dPCR quantification results from each triplicate were used to calculate the relative abundance of Zetaproteobacteria with regard to Bacteria.

## 16S rRNA gene sequencing and analyses

16S rRNA gene sequencing was performed at MR DNA (Shallowater, TX, USA) using the Illumina MiSeq technology in one sequencing run. Archaeal diversity was not assessed in this study as previous qPCR analyses of iron-rich microbial mats revealed that they represent a very small percentage of their microbial communities [39]. Only 16S rRNA genes of Bacteria were sequenced using 341F (5′ -CCTACGGGNGGCWGCAG) and 785R (5′ - GACTACHVGGGTATCTAATCC) primers targeting the V3-V4 hypervariable regions of the 16S rRNA gene [40] following the same procedures described in Astorch-Cardona et al. [41].

Data analysis was performed as described in Astorch-Cardona et al. [41] with trimming parameters set as follows: trimLeft = 17,21 and trimRight = 40. The Amplicon Sequence Variant (ASV) table was obtained following the process described in Astorch-Cardona et al. [41]. Unless otherwise specified, further sequence treatment was performed with the phyloseq package [42]. The classification of Zetaproteobacteria in ZetaOTUs was performed following the same specifications as described in Astorch-Cardona et al. [41].

The data sets generated and analyzed during the current study are publicly available. The data can be found here: NCBI, PRJNA1063670, the accession numbers for the BioSamples are SRR27489633—SRR27489647.

## Results

### Bacterial diversity in EMSO-LO samples

The results obtained from the 16S rRNA gene sequencing for Bacteria were of good quality, with a percentage of retained reads between 57.7% and 74.4% (S2 Table) of the total reads generated for each sample. Moreover, the rarefaction curves reached a plateau in all samples (S1 Fig), demonstrating that the sequencing effort was sufficient to evaluate the bacterial diversity in all of them.

Regarding alpha diversity, the Shannon index (Fig 3A) clearly showed that the three microbial colonizers containing either reduced or natural basalt presented a higher bacterial

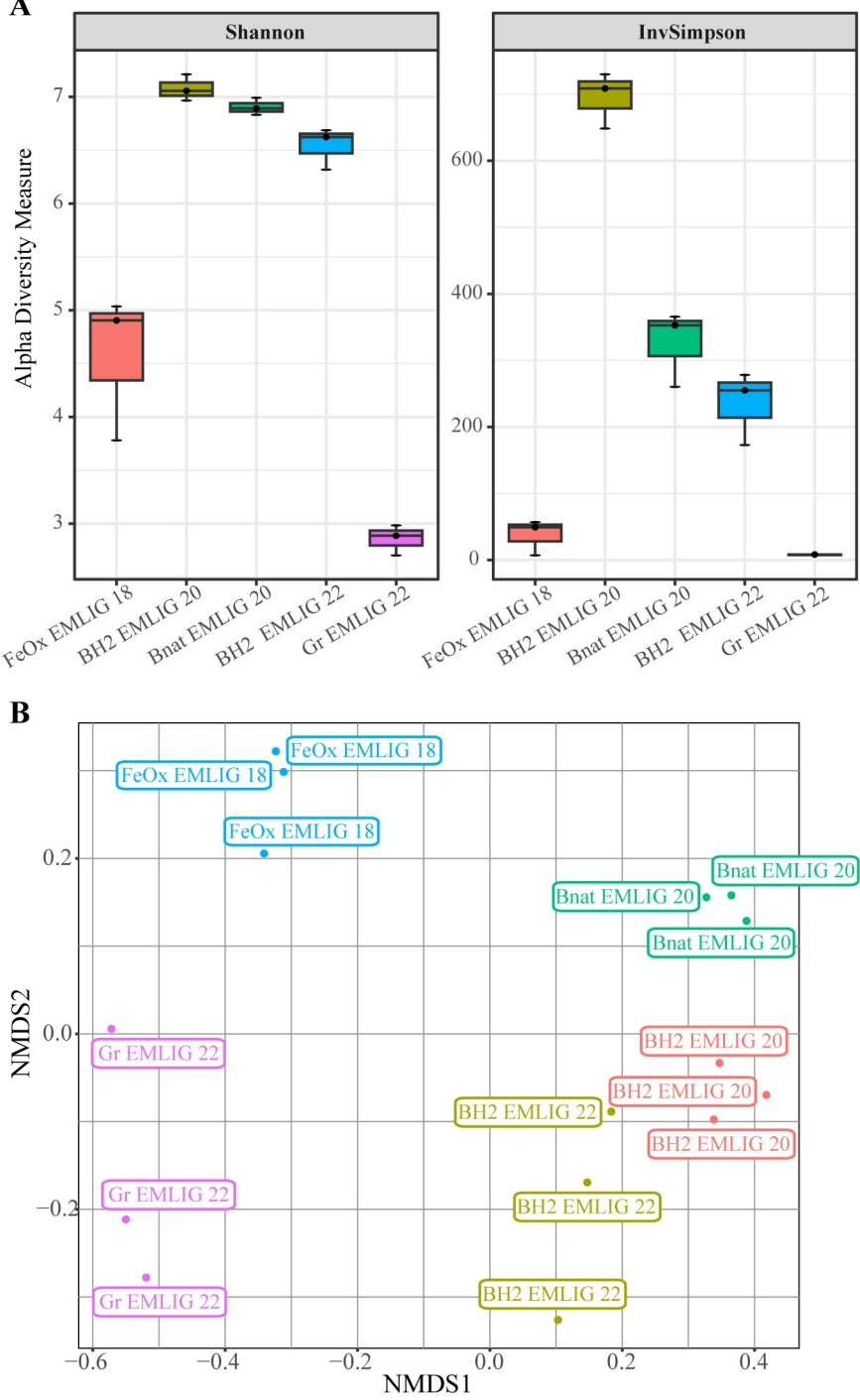

**Fig 3. Alpha and beta diversity of bacteria.** A) Box plot representing the Shannon and Inverse Simpson diversity indexes calculated for each bacterial community. B) NMDS plot representing the differences between the bacterial communities of all samples (stress = 0.08).

diversity than the iron-rich microbial mat and the microbial colonizer containing mild steel grit. Among the five samples, Gr EMLIG 22 was the one that presented the lowest alpha diversity index, while BH2 EMLIG 20 was the sample that had the highest one.

As for beta diversity, the NMDS plot revealed two main results (Fig 3B): (i) triplicates from each sample clustered together, corroborating the robustness of this analysis; and (ii) bacterial communities from each sample were significantly different from each other. This was confirmed by the PERMANOVA analysis by sample, which yielded a p-value of 0.001. Nonetheless, samples were mainly divided into two clusters regarding axis NMDS1: one formed by the FeOx EMLIG 18 and the Gr EMLIG 22 samples and the second formed by the BH2 EMLIG 20, Bnat EMLIG 20 and BH2 EMLIG 22 samples. The bacterial diversity between clusters was significantly different, as confirmed by both the PERMANOVA analysis for beta diversity (p-value 0.001) and the Wilcoxon Rank Sum Test for alpha diversity (p-value 0.001). In the first cluster, we could observe as well that the FeOx EMLIG 18 and the Gr EMLIG 22 samples were separated regarding axis NMDS2. Within the second cluster, axis NMDS2 separated samples containing natural basalt from those containing synthetic basalt. This analysis allowed us to discriminate the samples in an "anthropic" cluster including the FeOx EMLIG 18 and GR EMLIG 22 samples, and a "natural" cluster containing the BH2 EMLIG 20, Bnat EMLIG 20 and BH2 EMLIG 22 samples.

The bar plot (Fig 4) allowed us to investigate the differences between the bacterial communities of the samples more thoroughly. At the phylum level (Fig 4A), all the bacterial communities, except for those of the Gr EMLIG 22 sample and for one of the triplicates of the FeOx EMLIG 18 sample, were dominated by Proteobacteria (representing between 30.3% and 47.9% of the bacterial communities, S3 Table). Within Proteobacteria (Fig 4B), Gammaproteobacteria class dominated or were highly abundant in the microbial colonizers from the natural cluster (between 20.6–30.5%), and presented lower abundances in the two samples from the anthropic cluster (between 6.4–10.1%). Inversely, Alphaproteobacteria class dominated in the two anthropic cluster samples (16.3–28%) and presented non-negligible abundances in the samples from the natural cluster (12.9–25.8%). Regarding Zetaproteobacteria, the only sample in which they presented relative abundances of >1% was the FeOx EMLIG 18 iron-rich mat sample.

As already revealed on the NMDS plot regarding NMDS2 axis (Fig 3B), the two samples from the anthropic cluster hosted differing bacterial communities. The Gr EMLIG 22 sample was the only one dominated by the Campylobacterota (47.4%– 52.4%). Besides, it was characterized by a non-negligible abundance of members of the Bacteroidota (6.7–7.9) and the Firmicutes (2.4–3.9) phyla. Regarding the FeOx EMLIG 18 sample, the second most abundant phylum was the Bacteroidota (11.1–18.3%), followed by high abundances of members of the Patescibacteria (9.5–15.1%) and the Desulfobacterota (4–7%).

In comparison, the samples forming the natural cluster presented much more similar bacterial communities between them at the phylum level. Besides Proteobacteria and Bacteroidota (4.1–13.8%), they presented high abundances of members of the Planctomycetota (8.9–13.9%) and Actinobacteriota (4.2–10.6%) phyla. On the other hand, they were distinguished by the presence of the following group of phyla: Acidobacteriota (3–7.5%), NB1-j (2.8–6.5%), Verrucomicrobiota (2.4–4.5%), Gemmatimonadota (1.8–3.6%), Chloroflexi (1.1–4.5%), Myxococcota (0.7–2%), Nitrospirota (0.6–2.2%), Latescibacteria (0.3–1.4%) and Hydrogenedentes (0.2–1.1%).

## Zetaproteobacterial diversity in EMLIG samples

The quantification of Zetaproteobacteria (number of copies/μl of DNA) by dPCR was consistent and did not present high variations between triplicates (Fig 5A). Because we used different

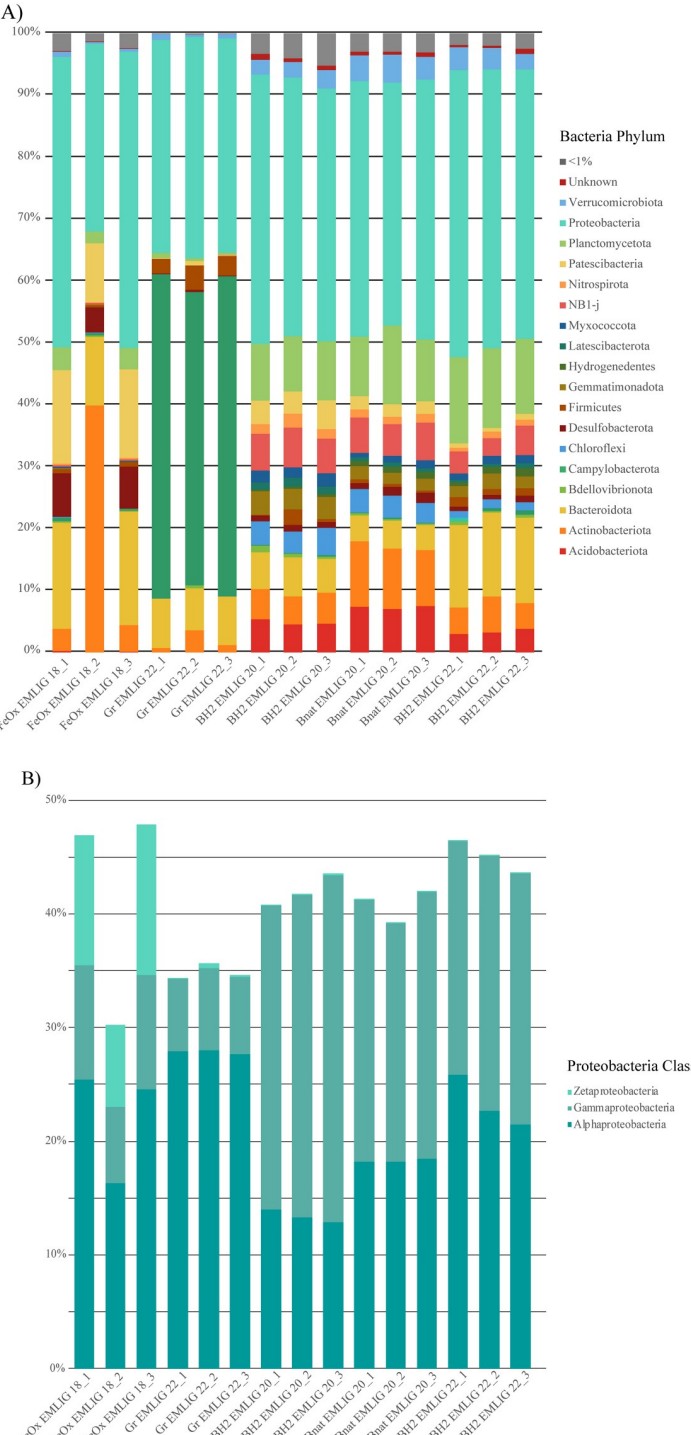

**Fig 4. Bacterial bar plot.** A) Bar plot depicting the relative abundance (%) of bacterial phyla for each triplicate of each sample. Only the phyla having an incidence higher than 1% are represented in the plot. B) Bar plot representing the relative abundance (%) of classes belonging to the Proteobacteria phylum.

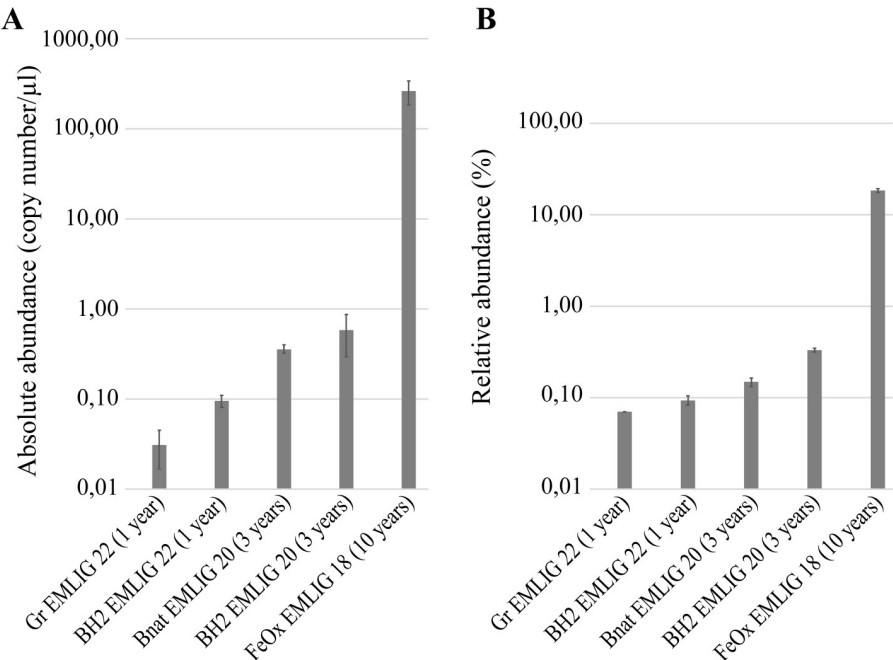

**Fig 5. Zetaproteobacterial quantification by dPCR.** A) Bar plot representing the absolute quantification of Zetaproteobacteria as the number of zetaproteobacterial 16S rRNA gene copies per µl of DNA sample. B) Bar plot representing the proportion of Zetaproteobacteria *vs* the total number of Bacteria.

dilution to analyze the samples, we tested the putative effect of the dilution and we observed no significant difference (Kruskal-Wallis test with a p-value <0.05). The results showed that Zetaproteobacteria were only found in higher abundances (261±77 copies/µl) in the FeOx EMLIG 18 iron-rich microbial mat sample than in the colonizers. The high sensitivity of dPCR allowed us to observe differences even between microbial colonizers (Fig 5A). Within them, Zetaproteobacteria presented higher abundances in the colonizers that had been deployed for a longer time, and in those that had an Fe(II)-enriched substratum (0.58±0.29 copies /µl for BH2 EMLIG 20 and 0.36±0.04 copies/µl for Bnat EMLIG 20) (Fig 5). In the colonizers that had been deployed for a shorter time, they presented higher abundances in basalt than in mild steel grit (0.09±0.01 copies/µl for BH2 EMLIG 22 and 0.03±0.01 copies/µl for Gr EMLIG 22). dPCR results were also used to calculate the relative abundance (%) of Zetaproteobacteria *vs* Bacteria in the samples (Fig 5B). The percentage of Zetaproteobacteria was higher in the FeOx EMLIG 18 iron-rich microbial mat sample than in the colonizers, where they represented <1% of the bacterial community, in agreement with the above presented data from 16S metabarcoding.

Zetaproteobacterial diversity was characterized *via* the ZetaOTUs classification. Within our dataset, there were 39 ASVs assigned to this class, which were further classified into 10 ZetaOTUs. The heatmap (Fig 6) allowed us to study how these ZetaOTUs were distributed among samples. Fig 6 revealed a higher zetaproteobacterial diversity in the anthropic cluster than in the natural one, contrary to the results obtained for bacterial diversity. Only one ZetaOTU was present in all studied samples: ZetaOTU 2. ZetaOTU 6 was present in all samples except for BH2 EMLIG 20. The iron-rich microbial mat sample harbored all ZetaOTUs found in the samples of this study, except for ZetaOTU 14, which was only present in the BH2 EMLIG 20 microbial colonizer. The Bnat EMLIG 20 and BH2 EMLIG 22 colonizers only contained ZetaOTUs 2 and 6.

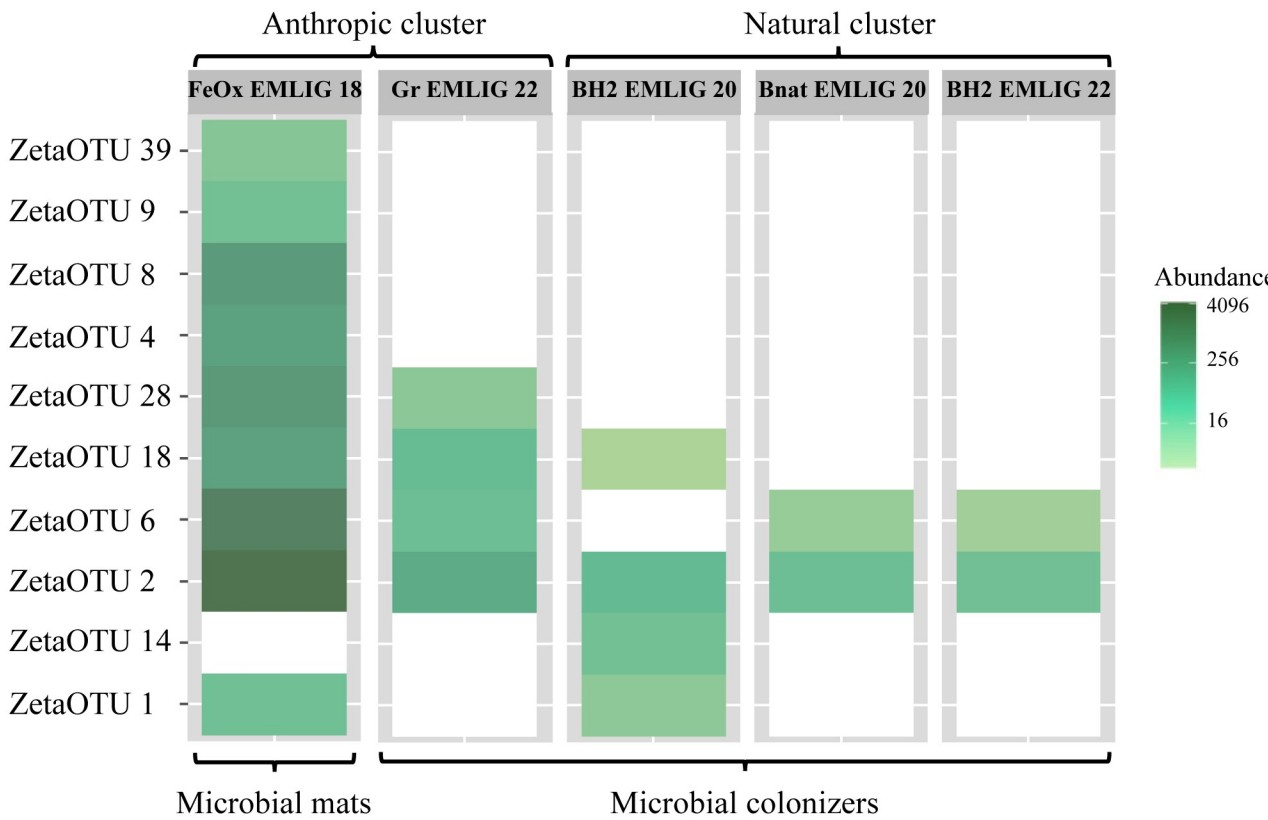

**Fig 6. Zetaprotobacteria heatmap.** Heatmap representing the presence/absence and abundance of each ZetaOTU in each sample. To construct the heatmap, ZetaOTUs were organized using an NMDS ordination and the Bray-Curtis dissimilarity index.

## Discussion

In marine habitats, established/mature iron-rich microbial mats have mainly been reported in iron-rich hydrothermal systems [43–46]. Nonetheless, they have also been described associated with shipwrecks and permanently submerged steel structures, in the form of rust, rusticles or tubercles [21–23]. In the Mediterranean Sea, these mats have only been reported in its continental margins' sediments, at depths between 300 and 800 m [47]. The present study is the first to report the presence of iron-rich microbial mats developing in anthropic Fe-rich substrata in the deep Mediterranean Sea (2400 m), a non-hydrothermal environment where dFe concentrations are naturally low (around 0.3–0.5 nM) [27]. The results from our previous studies of iron-rich mats in hydrothermal contexts [39, 41] revealed that their development and microbial communities structure depend on both the substratum type and the chemical composition and variability of their immediate environment [39].

### Effect of substratum type on bacterial diversity

This study allowed us to define two clusters regarding bacterial communities: an "anthropic" cluster including the FeOx EMLIG 18 and Gr EMLIG 22 samples, and a "natural" cluster containing the BH2 EMLIG 20, Bnat EMLIG 20 and BH2 EMLIG 22 samples. Despite the differences between clusters, some bacterial taxa were shared among the samples (Fig 4A), such as the typical bacterial phyla from marine sediments. These are usually dominated by the

Proteobacteria (including members of the Alpha- and Gammaproteobacteria classes), Bacteroidota, Chlamydiota, Firmicutes, Chloroflexi, Gemmatimonadota and Planctomycetota phyla [48, 49]. Except for the Chlamydiota, members of all the other phyla were present within the samples of this study. In fact, the deep coastal plain of the Mediterranean Sea is a sedimented environment, and as it has already been proposed in similar *in situ* incubations performed with mild steel [19], some of the bacterial populations colonizing the substrata most likely originate from the local marine sediments.

The bacterial communities of the anthropic cluster (Fig 4) differed from those of the natural cluster especially by the presence of Desulfobacterota and/or Campylobacterota. Members of these phyla are mainly involved in the sulphur cycle and are usually main players in microbially influenced corrosion [17]. Indeed, sulphate-reducing bacteria (*i.e.* several members of the Desulfobacterota phylum), use the sulphate present in seawater, producing sulphide and other corrosive metabolites that promote the process of corrosion [50] and that can be used by sulphur-oxidizing microorganisms for growth. Despite forming the anthropic cluster, the bacterial communities of these two samples largely differed from one another. One of the main differences was the dominance of Campylobacterota in the Gr EMLIG 22 sample. The most abundant ASVs belonging to this phylum (S4 Table) were identified as members of the *Sulfurimonas* and *Sulfurovum* genus, which are mainly sulphur- and hydrogen- oxidizing bacteria [51–54]. Barco et al. [13] identified Sulfurimonas as one of the most abundant taxa on mild steel incubated *in situ* both in marine surface sediments and in the water column. Metaproteomics on the same samples identified hydrogenases from Sulfurimonas species which can be related to the hydrogen metabolism [13], in relation with $H_2$ generation from iron corrosion in seawater. Moreover, this microbial group is usually present during early phases of colonization in microbially influenced corrosion [17], explaining their high abundance only in the Gr EMLIG 22 sample, which was incubated for less than one year. Regarding the FeOx EMLIG 18 sample, it was characterized by the presence of members of the Desulfobacterota phylum. Among them, we detected members of the *Desulfovibrio* genus, which has already been described as one of the biggest contributors to biocorrosion [23, 55, 56]. Other abundant ASVs (S4 Table) were related to methane oxidation (*i.e.* members of the *Methyloprofundus* genus or the *Marine Methylotrophic Groups 2* and *3*). Previous studies have revealed that both sulphate-reducing bacteria (*i.e.* members of the Desulfobacterota) and methanogen archaea play a very important role in later phases of microbially influenced corrosion [17]. Besides not having analyzed the archaeal communities, methanogen archaea could indeed be producing the methane necessary to support the development of the detected methanotrophic bacteria. Overall, our results revealed that these two samples contain bacterial communities involved in microbially influenced corrosion, but at different stages of development, explaining the differences observed among them. Indeed, the community developing on the mild steel grit (Gr EMLIG 22, one year of deployment) corresponds to an early phase of corrosion, while the mat developing on the SJB (FeOx EMLIG 18, 10 years of deployment) corresponds to a later phase. The biofilm on the Gr EMLIG 22 sample is thin enough to allow oxygen to penetrate the bacterial mat, allowing Campylobacterota to grow. When the community develops and matures with time, the bacterial mats become thicker, preventing oxygen to penetrate and therefore creating anoxic micro-niches that allow the development of anaerobic microorganisms such as sulphate-reducing bacteria or methanogen archaea that could sustain methanotrophic bacteria. Indeed, this scenario corroborates the model proposed by Emerson in 2018 [17].

The bacterial communities of the natural cluster (Fig 4) were characterized by the presence and non-negligible abundance of bacterial taxa that are typically found in seafloor basalt communities such as the Planctomycetota, Actinobacteriota, Bacteroidota, Acidobacteriota, Gemmatimonadota, Nitrospirota and Verrucomicrobiota phyla [7, 57–59]. In fact, basalt alteration

has been proven to provide sufficient energy for chemolithoautotrophic growth [7, 60], which seems to sustain very diverse but specific microbial communities. Interestingly, we observed a clear resemblance of the bacterial communities from the natural cluster to those of the iron-rich microbial mat from Lava Lake at the LSHF [39]. The iron-rich mats at Lava Lake develop with basalt as a substratum, which is very similar to that used for the microbial colonizers of the natural cluster.

As previously mentioned, environmental conditions remain mainly stable in the Mediterranean Sea [61], confirming that the differences between the bacterial communities of the anthropic and natural clusters can only be linked to the type of substratum used for this study. Nonetheless, it is important to bear in mind the difference that exists between the bacterial communities of the iron-rich mat and the microbial colonizer within the anthropic cluster, which reveals that incubation time and substratum composition are also key in the communities' structure. These results show therefore that at EMSO-LO, substratum type plays a major role in the bacterial communities associated with Fe-bearing materials.

## Zetaproteobacterial abundance linked to incubation time

Our data showed that Zetaproteobacteria only presented high abundances in the bacterial community of the iron-rich microbial mat sample, while its abundance was <1% in the microbial colonizers. Even though real-time quantitative PCR is the gold-standard method for gene detection and quantification, dPCR presents two main advantages: (i) an absolute quantification of the target DNA without the need of an external reference [62] and (ii) quantification of low abundance targets. This approach allowed us to quantify low zetaproteobacterial abundances in our samples. Moreover, quantification reproducibility among triplicates was optimal in all samples, making it an ideal technique for complex and low DNA-yielding samples such as those from microbial colonizers. In our study, we sampled different stages of microbial mats development; microbial colonizers were incubated *in situ* for either one or three years, representing intermediate stages of development, and the FeOx EMLIG 18 sample was collected after 10 years, representing a mature microbial mat. Our results indicate that Zetaproteobacteria present low abundances after one year of incubation that increase after 3 years and even more after 10 years (Fig 5). Unfortunately, earlier stages of colonization (*i.e.* the first weeks or months) could not be sampled in this study, but precedent studies revealed that Zetaproteobacteria are most probably early colonizers in such environments, using the Fe(II) released from steel or Fe-rich minerals for growth, suggesting that they create optimal niches for the development of other microorganisms and afterwards decrease in abundance when the conditions become more reducing [13, 18–20, 25]. Indeed, this could explain why their abundance is so low in our microbial colonizer samples.

Previous studies performed at intermediate stages of development have only been performed using Fe-rich minerals, and corroborate the low abundances of Zetaproteobacteria [63], except for a study performed in the Atlantic abyssal plain [15]. Nonetheless, such location presents different environmental and geological conditions (*e.g.* lower temperatures—around 2ºC), which could explain the disparity in the abundance of Zetaproteobacteria. Even though we did not analyze a mature iron-rich mat developing on basalt in this study, high abundances of Zetaproteobacteria in mats developing on basalts in hydrothermal ecosystems have been reported [41, 64, 65]. Analyses of microbially influenced corrosion from more long-term corrosion products, such as rusticles and tubercles from the World War-era II ship wrecks, or rust formed on mild steel emplaced for eight years in coastal sediments, have revealed the presence of low abundances of Zetaproteobacteria (maximum 3%) in their microbial communities [21–23], while in the FeOx EMLIG 18 sample they can represent up to a 19.3% of the bacterial

communities. This difference in abundance could be explained by the differing chemical composition of the substrata in which these mats have developed.

## Zetaproteobacterial diversity linked to both substratum type and incubation time

Our zetaproteobacterial diversity analysis revealed that ZetaOTU 2 was the only ZetaOTU that was present in all the samples of this study and was also the most abundant in all of them (Fig 6). This ZetaOTU is considered as a cosmopolitan ZetaOTU [66], and it has already been detected in previous incubation studies [4, 9, 18, 67]. To date, no strains of this ZetaOTU have been isolated. It is one of the ZetaOTUs forming the zetaproteobacterial core microbiome of iron-rich mats from the LSHF, together with ZetaOTUs 1, 4 and 17 [41]. Here, we corroborate that ZetaOTU 2 is cosmopolitan as well in the deep coastal plain at the Mediterranean Sea regardless of the substratum, suggesting that this ZetaOTU is critical for the development of these microbial communities, both in hydrothermal and non-hydrothermal contexts. This is the first time that ZetaOTUs 1 and 28 are reported in *in situ* incubation experiments. ZetaOTU 1 is part of the zetaproteobacterial core microbiome of the mats from the LSHF [41], and therefore its presence in different samples indicates its importance within both microbially influenced corrosion and mineral weathering. ZetaOTU 28 was the only one to be present exclusively in the two anthropic substrata. Regarding ZetaOTUs 6 and 18, they have been previously found in both mineral weathering [4, 13] and metal corrosion [4] incubations, with ZetaOTU 18 being more characteristic from microbially influenced corrosion.

Zetaproteobacterial diversity highlights an important difference between mature iron-rich microbial mats and microbial colonizers (Fig 6). The FeOx EMLIG 18 sample presented the highest diversity of ZetaOTUs, suggesting that Zetaproteobacteria diversify with time. Among the ZetaOTUs of this sample we could find ZetaOTUs 9 and 18, which are, up until now, the most characteristic of microbially influenced corrosion, as they have been detected in high abundances in different mild steel incubation studies [13, 16, 18]. It should be noted that these two ZetaOTUs have cultured representatives, *Ghiorsea bivora* TAG-1 and SV-108 for ZetaOTU9 [68], and *Mariprofundus. aestuarium* CP-5, *M. micogutta* ET2 and *Mariprofundus* sp. DIS-1 for ZetaOTU18 [25, 69, 70]. *G. bivora* is the only Zetaproteobacteria isolated to date, that has the ability to oxidize molecular hydrogen in addition to ferrous iron [68]. The absence of ZetaOTU 9 in all the microbial colonizers seemed controversial. This ZetaOTU has been described as the sole or as one of the most abundant ZetaOTUs in both metal corrosion and mineral weathering incubations [4, 15]. This could indeed be related to their capability to use hydrogen as electron donor [68]. Our results seem to indicate that ZetaOTU9 is only present in mature iron-rich mats, in the environmental conditions of the EMSO-LO area. Besides, the zetaproteobacterial communities of the FeOx EMLIG 18 sample also harbored some unique ZetaOTUs that had not previously been reported in such incubations: ZetaOTUs 4, 8 and 39. ZetaOTU 4 was also one of the ZetaOTUs forming the zetaproteobacterial core microbiome of the LSHF [41]. The presence of ZetaOTU 4 in the only sample that consists of an iron-rich microbial mat and its absence in all the microbial colonizers could indicate that this ZetaOTU is essential for the development of mature microbial mats. Zetaproteobacterial diversification with time is indeed corroborated by the incubation experiments using the BH2 substratum, which allow us to compare the communities that developed either after one or three years.

Finally, it is important to highlight that the Bnat EMLIG 20 and the BH2 EMLIG 22 zetaproteobacterial communities were equal (Fig 6), despite containing different substrata and having been incubated for different time-periods. Indeed, such a low diversity of ZetaOTUs

in the Bnat EMLIG 20 sample could be explained by the lower abundance of Fe(II) in the natural basalt substratum, which could be therefore a selective pressure for certain ZetaOTUs.

## Supporting information

**S1 Fig. Rarefaction curves for bacteria.** Rarefaction curves for Bacteria for each triplicate of each sample.
(PDF)

**S1 Table. dPCR primers.** Primers used for dPCR with respective annealing temperature and cycle number for denaturation step.
(PDF)

**S2 Table. Sequencing and sequence curation information.** Number of reads after each step of the DADA2 pipeline and percentage of retained reads after treatment.
(PDF)

**S3 Table. Bacterial phyla abundance.** Abundance of the different bacterial phyla for each sample.
(XLSX)

**S4 Table. 100 most abundant bacterial ASVs.** List of the 100 most abundant bacterial ASVs, their corresponding genera, NCBI accession number of their closest relative (isolate or strain), and percentage of sequence identity.
(XLSX)

## Acknowledgments

We would like to thank the chief scientists of the different campaigns, as well as the scientific team, sailors, and underwater gear personnel of the French Oceanographic Fleet for sample collection. We thank the captains, officers, and crew onboard R.V. *PourquoiPas*? who made the series of EMSO Ligure Ouest cruises possible. We thank the ROV *Victor 6000* and HOV *Nautile* team for supporting our deep submergence field campaigns.

## Author Contributions

**Conceptualization:** Lionel Bertaux, Yann Denis, Alain Dolla, Céline Rommevaux.

**Data curation:** Aina Astorch-Cardona, Lionel Bertaux.

**Formal analysis:** Aina Astorch-Cardona, Lionel Bertaux, Yann Denis, Alain Dolla, Céline Rommevaux.

**Funding acquisition:** Céline Rommevaux.

**Methodology:** Aina Astorch-Cardona, Lionel Bertaux, Yann Denis, Alain Dolla, Céline Rommevaux.

**Project administration:** Céline Rommevaux.

**Resources:** Aina Astorch-Cardona, Céline Rommevaux.

**Supervision:** Yann Denis, Alain Dolla, Céline Rommevaux.

**Validation:** Aina Astorch-Cardona, Lionel Bertaux, Yann Denis, Alain Dolla, Céline Rommevaux.

**Visualization:** Aina Astorch-Cardona, Céline Rommevaux.

**Writing – original draft:** Aina Astorch-Cardona.

**Writing – review & editing:** Aina Astorch-Cardona, Lionel Bertaux, Yann Denis, Alain Dolla, Céline Rommevaux.

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
