## [Decision Letter · Decision Letter 0]

7 May 2024

PONE-D-24-02517Bacterial and zetaproteobacterial communities from iron-rich microbial mats and microbial colonizers in the Mediterranean Sea (EMSO-Western Ligurian Sea Observatory)PLOS ONE

Dear Dr. ROMMEVAUX,

Thank you for submitting your manuscript to PLOS ONE. After careful consideration, we feel that it has merit but does not fully meet PLOS ONE’s publication criteria as it currently stands. Therefore, we invite you to submit a revised version of the manuscript that addresses the points raised during the review process.

Sorry for the time it took; your manuscript has been evaluated by two independent reviewers and both agree this is a very interesting study and worth to publish; yet they have some comments/questions that I would kindly ask you to address/reply

We look forward to receiving your revised manuscript.

Kind regards,

Clara F. Rodrigues

Academic Editor

PLOS ONE

 [Flotte Océanographique Française (FOF) Grant EMSO-LO cruises 2018, 2022

Agence Nationale de la Recherche (ANR) Grant ANR-21- CE02-0012

Ministère de l'Education Nationale, de l'Enseignement Superieur et de la Recherche (MESR)  Grant Astorch-Cardona PhD scholarship

EC European Regional Development Fund(ERDF) grant 1166-39417].  

[We would like to thank the chief scientists of the different campaigns, as well as the scientific team, sailors, and underwater gear personnel of the French Oceanographic Fleet for sample collection. We thank the captains, officers, and crew onboard R.V. PourquoiPas? who made the series of EMSO Ligure Ouest cruises possible. We thank the ROV Victor 6000 and HOV Nautile team for supporting our deep submergence field campaigns.

This project was funded by the French Oceanographic Fleet through the 2018, 2020 and 2022 cruises within the EMSO Ligure Ouest program (France). This work was supported by a French National Research Agency (ANR) grant as part of the IRONWOMAN project (ANR-21-CE02-0012). A.A.C. was supported by an MESR PhD scholarship and Aix-Marseille University (France). The project leading to this publication has received funding from the European FEDER Fund under project 1166-39417. ]

 [Flotte Océanographique Française (FOF) Grant EMSO-LO cruises 2018, 2022

Agence Nationale de la Recherche (ANR) Grant ANR-21- CE02-0012

Ministère de l'Education Nationale, de l'Enseignement Superieur et de la Recherche (MESR)  Grant Astorch-Cardona PhD scholarship

EC European Regional Development Fund(ERDF) grant 1166-39417]. 

6. We note that Figure 2 and 3 in your submission contain copyrighted images. All PLOS content is published under the Creative Commons Attribution License (CC BY 4.0), which means that the manuscript, images, and Supporting Information files will be freely available online, and any third party is permitted to access, download, copy, distribute, and use these materials in any way, even commercially, with proper attribution. For more information, see our copyright guidelines: http://journals.plos.org/plosone/s/licenses-and-copyright.

a. You may seek permission from the original copyright holder of Figure 2 and 3 to publish the content specifically under the CC BY 4.0 license. 

7. We note that Figure 1 in your submission contain [map/satellite] images which may be copyrighted. All PLOS content is published under the Creative Commons Attribution License (CC BY 4.0), which means that the manuscript, images, and Supporting Information files will be freely available online, and any third party is permitted to access, download, copy, distribute, and use these materials in any way, even commercially, with proper attribution. For these reasons, we cannot publish previously copyrighted maps or satellite images created using proprietary data, such as Google software (Google Maps, Street View, and Earth). For more information, see our copyright guidelines: http://journals.plos.org/plosone/s/licenses-and-copyright.

Reviewers' comments:

Reviewer's Responses to Questions

**Comments to the Author**

1. Is the manuscript technically sound, and do the data support the conclusions?

Reviewer #1: Yes

Reviewer #2: Yes

2. Has the statistical analysis been performed appropriately and rigorously? 

Reviewer #1: Yes

Reviewer #2: Yes

3. Have the authors made all data underlying the findings in their manuscript fully available?

Reviewer #1: Yes

Reviewer #2: Yes

4. Is the manuscript presented in an intelligible fashion and written in standard English?

Reviewer #1: Yes

Reviewer #2: Yes

5. Review Comments to the Author

Reviewer #1: This manuscript highlights the SSU amplicon sequencing and dPCR assessment of iron-rich microbial mats and microbial colonizers found at the EMSO-Western Ligurian Sea Observatory with a focus on Zetaproteobacteria. Itemized specific comments:

(1) In the introduction there are several references describing the occurrence of Zetaproteobacteria and various ecosystems, like hydrothermal vents. No included are where microbial growth chambers have been used at hydrothermal vents across the Pacific showing distinct biogeographic signatures of autotrophic communities including Zetaproteobacteria as the initial colonizers at many sites (Fullerton et al., 2024; doi 10.1093/femsec/fiae001). Other habitats like oligotrophic freshwater lakes have included Zetaproteobacteria, like where Zeta OTUs 2 and 6 have also been detected in the benthic hot spring mats at Crater Lake, OR (Stromecki et al., 2022; doi 10.3389/fmicb.2022.876044). Also see lines 343 to 344 and 446 to 448 in the discussion section.

(2) In the introduction (line 94), where EMSO is used for the first time, this should be spelled out for readers not familiar with this abbreviation.

(3) For Figure 1 and 2, can this map (Fig 1) and these photos (Fig 2) be made with more resolution, they seem very pixilated.

(4) It seems rather hard to make any conclusions regarding colonization from colonization modules that were deployed for ~1 year (line 183), how do you reconcile this with your conclusions? Also see lines 435 to 440 in the discussion section.

(5) Regarding dPCR, in line 216, you state that 45 to 75 cycles were used. This seems rather excessive. Can you show any evidence that detections were made in prior to ~25-30 cycles?

(6) Line 219, the word “pression” should read as “pressure”.

(7) Line 429, change “between” to “among”.

(8) Line 440, period at the end of sentence here.

Reviewer #2: This is a review of the manuscript titled “Bacterial and zetaproteobacterial communities from iron-rich microbial mats and microbial colonizers in the Mediterranean Sea (EMSO-Western Ligurian Sea Observatory) by Astorch-Cardona et al. This is a study that analyzed the colonization of various Fe-bearing substrates including basalt and mild steel by bacteria. This work builds on the foundation laid by other recent studies on basalt and mild steel and corroborates for the most part the results of those studies. The value of the paper is in the different geographical and environmental context, with in-situ incubations done in the Mediterranean with Fe levels much lower than in other sites tested (i.e. Juan de Fuca) or at lower temperatures (2 to 4C), compared to shallower marine sites. So, this study is able to corroborate most of those results but also provides some new findings specific to the Zetaproteobacteria.

Although, the paper is overall well-written; there are some areas that need some polishing. Some major/minor comments are included below.

It’s not clear why the Archaea are not included in the analysis. What is the reason? I would also favor replacing Figure 6 with Table S3 in the main text, which clearly shows the values of the abundances of Zetas near 0 in most of the samples. This is hard to see in Figure 5, and the fact that the symbol is shown for the Zetaproteobacteria in the figure is confusing. I would suggest removing the Zeta symbol in Figure 5 (for samples where the abundance is near 0) because it should be accounted for in the “<1%” category. By doing so, and adding Table S3, the reader will have all the important information in clear sight. I suggest moving Figure 6 to the Supplemental section.

L1: The title is somewhat redundant. The word “Bacterial” necessarily includes the Zetaproteobacteria. The word “microbial” is used twice.

L49-50: These lines are verbatim L27-29 in the abstract.

L51-54: These lines are verbatim L29-32.

L55: I believe Dave Karl proposed that concept earlier in 1988 (Loihi Seamount discovery; Nature paper). In that paper he presents evidence.

L72: Reference 10 is more appropriate here; it’s a 2011 paper. Reference 9 is a 2018 paper.

L155: What type of steel is the steel grit used in this study? It is not clear if it is stainless steel or mild steel. Please specify.

L330: Indicate what ASV means here.

L339: Change “Zetaprotobacterial” to “Zetaproteobacteria”.

L373: Is this meant to be stainless steel or mild steel. Also, basalt does contain sulfur, at low % but comparable to mild steel.

L377-379: Sulfurimonas and Sulfurovum can also oxidize hydrogen. Steel produces H2 when it comes in contact with seawater. Barco et al. (reference 14) identified Epsilonproteobacteria on steel as well as hydrogenase proteins belonging to Epsilonproteobacteria on this substrate. This possibility should be discussed in this section. Mori et al. (2017; not cited in the manuscript) also discusses H2 oxidation in Zetaproteobacteria, more specifically ZetaOTU9 (Ghiorsea bivora). Since ZetaOTU 9 is seen in the microbial mat, growing on steel, this possibility should be discussed as well.

L448: This sentence seems unfinished.

L457: Is there an isolated Zeta from ZetaOTU 2? Please specify in the manuscript. Also, what are the Zeta OTUs linked to the different isolated Mariprofundus species? Specifying this in the main text will help interpret some of the data.

6. PLOS authors have the option to publish the peer review history of their article (what does this mean?). If published, this will include your full peer review and any attached files.

Reviewer #1: **Yes: **Craig Lee Moyer

Reviewer #2: No

---

## [Author Response · Author response to Decision Letter 0]

31 May 2024

RESPONSE TO REVIEWER COMMENTS

We would like to thank the reviewers for their comments and useful suggestions on our manuscript, which have allowed us to revise and substantially improve it. Hereafter you will find a detailed response for each of the reviewers’ comments. The pages and line numbers indicated are those of the revised manuscript without marked corrections.

Reviewer #1: 

 (1) In the introduction there are several references describing the occurrence of Zetaproteobacteria and various ecosystems, like hydrothermal vents. No included are where microbial growth chambers have been used at hydrothermal vents across the Pacific showing distinct biogeographic signatures of autotrophic communities including Zetaproteobacteria as the initial colonizers at many sites (Fullerton et al., 2024; doi 10.1093/femsec/fiae001). Other habitats like oligotrophic freshwater lakes have included Zetaproteobacteria, like where Zeta OTUs 2 and 6 have also been detected in the benthic hot spring mats at Crater Lake, OR (Stromecki et al., 2022; doi 10.3389/fmicb.2022.876044). Also see lines 343 to 344 and 446 to 448 in the discussion section.

We agree with the reviewer in this comment, and we have modified the introduction section to add the Fullerton (line 55, p. 3) and Stromeki (line 49, p. 3) references, introducing thus other environments than marine ones and previous studies on colonizers (lines 54-55, p.3).

 (2) In the introduction (Line 94), where EMSO is used for the first time, this should be spelled out for readers not familiar with this abbreviation.

EMSO has been spelled out (line 95-96, p. 5).

(3) For Figure 1 and 2, can this map (Fig 1) and these photos (Fig 2) be made with more resolution, they seem very pixilated.

Figure 1 has been deleted in the revised version and replaced by the GPS coordinates of each site in the text (lines 124 and 126, p. 6). The resolution of Figure 2 (now Figure 1 in the revised version) has been improved to match the quality standard of the journal.

 (4) It seems rather hard to make any conclusions regarding colonization from colonization modules that were deployed for ~1 year (line 183), how do you reconcile this with your conclusions? Also see lines 435 to 440 in the discussion section.

We do not base our conclusions on the analyses of the colonizers deployed only for one year but rather on the comparison of Zetaproteobacteria abundances on colonizers deployed for 1 year, 3 years and on a mature mat established for 10 years. To make this point clearer we have modified the text in the discussion section (line 437, p. 19). In addition, to illustrate this purpose, we have changed the order of samples in Figure 5 from the shortest to the longest incubation time. 

(5) Regarding dPCR, in line 216, you state that 45 to 75 cycles were used. This seems rather excessive. Can you show any evidence that detections were made in prior to ~25-30 cycles?

We want to emphasize that we performed digital PCR (dPCR) instead of quantitative Real-Time PCR (qRT-PCR) to be able to absolutely quantify the bacteria. Statistically, a majority of positive droplets in a dPCR contain only one PCR target. This is why the number of cycles is higher than that usually found for qRT-PCR to reach the amplification plateau which is the analysis step. Usually, a minimum of 45 cycles is recommended by the supplier (Stilla Technologies). Our assays with only 30 cycles using dPCR gave too low signals to distinguish in between positive and negative droplets. In addition, it is known that increasing the number of cycles with dPCR allows to discriminate positive and negative droplets without increasing the aspecificity of the reaction, giving therefore robust and confident results as demonstrated by Witte et al. (2016) (Witte AK, Mester P, Fister S, Witte M, Schoder D, Rossmanith P (2016) A Systematic Investigation of Parameters Influencing Droplet Rain in the Listeria monocytogenes prfA Assay - Reduction of Ambiguous Results in ddPCR. PLoS ONE 11(12): e0168179. doi:10.1371/journal.pone.0168179).

 (6) Line 219, the word “pression” should read as “pressure”.

Done (line 218, p. 10).

 (7) Line 429, change “between” to “among”.

Done (line 431, p.18).

(8) Line 440, period at the end of sentence here.

Done (line 442, p. 19).

Reviewer #2: 

It’s not clear why the Archaea are not included in the analysis. What is the reason? 

Archaeal diversity was not assessed in this study as previous qRT-PCR analyses of iron-rich microbial mats revealed that they represented a very small percentage of their microbial communities (Astorch-Cardona et al 2023). We have added this sentence (lines 231-233, p. 10) to clarify this purpose. In addition, the present study focuses on Bacteria and specially on Zetaproteobacteria. 

I would also favor replacing Figure 6 with Table S3 in the main text, which clearly shows the values of the abundances of Zetas near 0 in most of the samples. This is hard to see in Figure 5, and the fact that the symbol is shown for the Zetaproteobacteria in the figure is confusing. I would suggest removing the Zeta symbol in Figure 5 (for samples where the abundance is near 0) because it should be accounted for in the “<1%” category. By doing so, and adding Table S3, the reader will have all the important information in clear sight. I suggest moving Figure 6 to the Supplemental section.

We would like to keep Figure 6 in the text because it is an important point in the paper, as it shows the absolute quantification of Zetaproteobacteria in the samples via dPCR. On the other hand, Table S3 shows the relative proportion of different phyla from metabarcoding analyses. The relative proportion of each class among the phylum Proteobacteria is indicated in the same table in the last three rows. In this table, the “<1%”category does not correspond to Zetaproteobacteria (which is a class within Proteobacteria phylum) but to the other the phyla with lower abundances than1% To avoid any confusion, we have modified Table S3 by adding phylum and classes in the first column. We agree with the reviewer that Figure 5 is not clear, and we have modified it by splitting it in two parts: Figure 5A describing the phyla diversity, and Figure 5B describing the diversity among the Proteobacteria phylum, and so emphasizing the relative proportion of Zetaproteobacteria in the samples.

L1: The title is somewhat redundant. The word “Bacterial” necessarily includes the Zetaproteobacteria. The word “microbial” is used twice.

We agree with the reviewer and we have changed the title (lines 1-3, p. 1)

L49-50: These lines are verbatim L27-29 in the abstract.

L51-54: These lines are verbatim L29-32.

We agree with the reviewer and have deleted these sentences, starting the abstract directly on Zetaproteobacteria (line 27, p. 2). Regarding this, we have also changed the first paragraph of the introduction (lines 45-55, p. 3).

L55: I believe Dave Karl proposed that concept earlier in 1988 (Loihi Seamount discovery; Nature paper). In that paper he presents evidence.

In their paper, Karl et al. (1988) describe the presence of iron-depositing bacteria, but without any mention about their metabolisms, and the way they can grow in this environment. We think that the concept of the involvement of neutrophilic iron-oxidizing bacteria in mineral weathering processes in marine environments was effectively introduced by Wirsen et al. (1993) in their paper. Therefore, we prefer to keep this sentence without making reference to Karl et al. (1988)..

L72: Reference 10 is more appropriate here; it’s a 2011 paper. Reference 9 is a 2018 paper.

We agree with the reviewer and have changed this reference by indicating 2011 paper before the 2018 paper in the text (lines 73 and 75, p. 4)

L155: What type of steel is the steel grit used in this study? It is not clear if it is stainless steel or mild steel. Please specify.

Steel from SJB structure serving as a support for iron-rich mat development, together with steel grit used in the colonizers are mild steel. We have specified the type of steel in lines 135 p. 6; 154, p. 7; 318, p. 14; 396, p. 17.

L330: Indicate what ASV means here.

Done (lines 238-239, p10-11).

L339: Change “Zetaprotobacterial” to “Zetaproteobacteria”.

Done (line 339, p. 15).

L373: Is this meant to be stainless steel or mild steel. Also, basalt does contain sulfur, at low % but comparable to mild steel.

We agree with the reviewer that MORB (Mid-Ocean Ridge Basalt) contained sulfur at low percentages, comparable to mild steel, so we deleted this sentence. (line 373, p. 16); we have also specified (see above) that for our study, we used mild steel and not stainless steel (lines 135 p. 6; 154, p. 7; 318, p. 14; 396, p. 17).

L377-379: Sulfurimonas and Sulfurovum can also oxidize hydrogen. Steel produces H2 when it comes in contact with seawater. Barco et al. (reference 14) identified Epsilonproteobacteria on steel as well as hydrogenase proteins belonging to Epsilonproteobacteria on this substrate. This possibility should be discussed in this section. Mori et al. (2017; not cited in the manuscript) also discusses H2 oxidation in Zetaproteobacteria, more specifically ZetaOTU9 (Ghiorsea bivora). Since ZetaOTU 9 is seen in the microbial mat, growing on steel, this possibility should be discussed as well.

We agree with the reviewer and have added “and hydrogen” to sulphur-oxidizing bacteria (line 377, p. 16)). In addition, we have added a sentence about the metaproteomic study on hydrogenases from Sulfurimonas (Barco et al., 2017), in relation with H2 generation from iron corrosion in seawater (lines 378-381, p. 16). 

L448: This sentence seems unfinished.

We have deleted the end of this sentence (line 451, p. 19).

L457: Is there an isolated Zeta from ZetaOTU 2? Please specify in the manuscript. Also, what are the Zeta OTUs linked to the different isolated Mariprofundus species? Specifying this in the main text will help interpret some of the data.

We have specified in the text that, to date, no strain of ZetaOTU2 have been isolated (lines 463-464, p. 20). , Only ZetaOTU9 and ZetaOTU18 have isolated strains and we have indicated it in the text (lines 481-484, p. 20). We also refer to Ghiorsea bivora, the only strain having the capability to oxidize hydrogen in addition to ferrous iron with the reference to Mori et al. (2017) (lines 484-485, p. 20-21).

We hope that our changes in this revised version answer the reviewers’ comments and that they will find the revised version suitable for publication now.

---

## [Editor Report · Decision Letter 1]

4 Jun 2024

Diversity and dynamics of Bacteria from iron-rich microbial mats and colonizers in the Mediterranean Sea (EMSO-Western Ligurian Sea Observatory): focus on Zetaproteobacteria

PONE-D-24-02517R1

Dear Dr. ROMMEVAUX,

We’re pleased to inform you that your manuscript has been judged scientifically suitable for publication and will be formally accepted for publication once it meets all outstanding technical requirements.

Kind regards,

Clara F. Rodrigues

Academic Editor

PLOS ONE

Additional Editor Comments (optional):

Thank you for addressing all the reviewers' questions and concerns.
---

## [Editor Report · Acceptance letter]

7 Jun 2024

PONE-D-24-02517R1 

PLOS ONE

Dear Dr. ROMMEVAUX, 

I'm pleased to inform you that your manuscript has been deemed suitable for publication in PLOS ONE. Congratulations! Your manuscript is now being handed over to our production team.

Kind regards, 

on behalf of

Dr. Clara F. Rodrigues 

Academic Editor

PLOS ONE